

# Evaluating probabilistic programming and fast variational Bayesian inference in phylogenetics

Mathieu Fourment and Aaron E. Darling

ithree Institute, University of Technology Sydney, Sydney, NSW, Australia

## ABSTRACT

Recent advances in statistical machine learning techniques have led to the creation of probabilistic programming frameworks. These frameworks enable probabilistic models to be rapidly prototyped and fit to data using scalable approximation methods such as variational inference. In this work, we explore the use of the Stan language for probabilistic programming in application to phylogenetic models. We show that many commonly used phylogenetic models including the general time reversible substitution model, rate heterogeneity among sites, and a range of coalescent models can be implemented using a probabilistic programming language. The posterior probability distributions obtained via the black box variational inference engine in Stan were compared to those obtained with reference implementations of Markov chain Monte Carlo (MCMC) for phylogenetic inference. We find that black box variational inference in Stan is less accurate than MCMC methods for phylogenetic models, but requires far less compute time. Finally, we evaluate a custom implementation of mean-field variational inference on the Jukes–Cantor substitution model and show that a specialized implementation of variational inference can be two orders of magnitude faster and more accurate than a general purpose probabilistic implementation.

## INTRODUCTION

Markov chain Monte Carlo (MCMC) algorithms have become the workhorse of Bayesian phylogenetic inference since they were introduced in the late 1990's (*Mau & Newton, 1997*; *Larget & Simon, 1999*). Recent advances in computing hardware and corresponding software implementations have allowed this class of inference method to handle increasingly large datasets (*Flouri et al., 2015*; *Ayres et al., 2019*). However the quantity of sequence data being generated every year has been growing exponentially, which, when combined with practitioner's desires to conduct inference on increasingly rich statistical models, makes MCMC algorithms difficult to apply in practice because they are too slow to compute. Sequential Monte Carlo is an alternative sampling method (*Doucet, De Freitas & Gordon, 2001*) that has recently received some attention in the phylogenetic community (*Bouchard-Côté, Sankararaman & Jordan, 2012*;

Corresponding author
Mathieu Fourment,
mathieu.fourment@uts.edu.au

*Wang, Bouchard-Côté & Doucet, 2015*; *Fourment et al., 2018a*). Although it is fast and, unlike MCMC algorithms, easily parallelizable, designing efficient proposals for tree topologies and continuous parameters has proven to be difficult (*Fourment et al., 2018a*). Unlike some statistical models, phylogenetic models have a structure that makes approximating their posterior distribution especially difficult. Specifically, the combination of dependencies among discrete-valued (e.g. the tree topology) and continuous variables (e.g. branch lengths, rates, etc.) means that many recently developed methods to accelerate Bayesian inference can not be directly applied to phylogenetic models.

## Attempts to accelerate MCMC

The tree topology is arguably the most difficult part of a phylogenetic model to estimate. The number of possible topologies for binary trees grows super-exponentially in the number of sequences (tips). As a result, sophisticated MCMC proposals are required to efficiently explore the enormous space of this parameter. Typically, MCMC samplers propose a new topology using simple rearrangements of the current topology such as nearest neighbor interchange and subtree-prune-regraft operations. However these relative small, local moves through the state space can lead to inadequate sampling, especially when the posterior is multimodal (*Whidden & Matsen, 2015*). Several advances in tree topology sampling have been proposed, such as the use of a conditional clade probability distribution to guide proposals (*Höhna & Drummond, 2011*). Another recently proposed approach involved an extension of Hamiltonian Monte Carlo, which uses gradient information to guide proposals, to work in the discrete state space of tree topologies (*Dinh et al., 2017*).

When it comes to continuous parameters, the simplest proposals update a single parameter at a time (e.g. via a multiplier or sliding window proposals). This comes at the expense of ignoring correlation among parameters. Building upon research on adaptive MCMC (*Haario, Saksman & Tamminen, 2001*; *Roberts & Rosenthal, 2009*), recent efforts have implemented multivariate proposals to update blocks of continuous phylogenetic model parameters (*Baele et al., 2017*). While *Baele et al. (2017)* showed that this proposal is statistically more efficient than standard proposals in terms of effective sampling size per unit of time, this proposal is currently not available for updating branch lengths in the BEAST packages (*Suchard et al., 2018*; *Bouckaert et al., 2019*). Finally, in order to increase the efficiency of an MCMC sampler, *Aberer, Stamatakis & Ronquist (2015)* developed an independence sampler to update branch lengths in an informed way. Unfortunately, none of these approaches provide the magnitude of improvement required to carry out Bayesian phylogenetic inference on the datasets of 1,000s of sequences which have now become commonplace.

## Variational inference: an alternative to MCMC

One alternative to MCMC that has been proposed for Bayesian inference of model parameters is variational Bayes (VB) (*Jordan et al., 1999*; *Wainwright & Jordan, 2008*). Like MCMC, VB is a technique used to approximate intractable integrals. The main idea behind variational inference is to transform the posterior approximation of an intractable

model into an optimization problem using a family of approximate densities that are tractable. The aim is to find the member of that family with the minimum Kullback–Leibler (KL) divergence to the posterior distribution of interest. Although variational inference does not provide the guarantee that MCMC would of generating samples from the true posterior distribution, variational inference tends to be much faster than MCMC since it relies on fast optimization methods. It is common for VB inference to require 100-fold less compute time than MCMC to approximate the same posterior distribution, although the quality of the approximation may not be as good. Variational inference has become popular in machine learning, as evidenced by the large number of software libraries implementing it (e.g. Stan (*Carpenter et al., 2017*), PyMC (*Salvatier, Wiecki & Fonnesbeck, 2016*), TensorFlow (*Abadi et al., 2015*), Edward (*Tran et al., 2016*)).

To date, VB has received relatively little attention in the field of phylogenetics, possibly due to the presence of discrete model components such as the tree topology which can not be inferred using standard VB algorithms. However, in recent years interest in VB for phylogenetics has begun to grow. Using the Jukes–Cantor (JC69) nucleotide substitution model and fixed topologies, *Fourment et al. (2018b)* employed VB to approximate the marginal likelihood of fixed topologies. *Zhang & Matsen (2019)* have recently shown that VB is a superior method to infer phylogenetic tree posteriors in the classical case of unrooted trees with branch lengths and the JC69 substitution model. *Dang & Kishino (2019)* used variational inference to approximate the CAT-Poisson model for amino acid data (*Lartillot & Philippe, 2004*). In that work the authors used a partial Gibbs sampling strategy to update the topology and therefore only the continuous parameters were approximated with VB.

In the present work, we describe implementations of several phylogenetic models for nucleotide datasets using the Stan language (*Carpenter et al., 2017*). These models are substantially richer than those described in previous studies of variational inference for phylogenetics as they include more general nucleotide substitution models, such as the general time reversible (GTR) substitution model, and rate heterogeneity across sites. We also implemented molecular clock and coalescent models to infer the substitution rate and divergence times of heterochronous and homochronous sequence data. As with previous studies, we consider the topology to be fixed (*Fourment et al., 2018b*). Despite this limitation, implementing the type of model we describe here under a fixed tree is a useful step forward toward fast Bayesian inference of complex phylogenetic models, as it helps us understand the quality and speed of posterior approximation that can be achieved using a generic modeling language like Stan. We compared the performance of the variational approximations to that obtained with the widely used MCMC-based phylogenetic software package BEAST, run on a fixed tree topology (*Suchard et al., 2018*; *Bouckaert et al., 2019*). Because our models are implemented in the Stan language, we are able to carry out inference using any of the inference engines available in Stan, and so we used the No-U-Turn Sampler (NUTS) (*Hoffman & Gelman, 2014*) to further validate our phylogenetic model implementations. Finally, we compared the mean-field approximations obtained with `phylostan` to those obtained with a highly specialized implementation of mean-field variational inference in the C language

**Table 1 Models implemented in phylostan.**

| Type | Model |
|------|-------|
| Substitution models | JC69, HKY, GTR |
| Rate heterogeniety across sites | Proportion of invariant sites, Weibull and discrete distributions |
| Clock model | Clock-free, strict, relaxed (hierarchical): autocorrelated, uncorrelated |
| Coalescent | Constant, skyride, skygrid |

software `physher` (*Fourment & Holmes, 2014*; *Fourment et al., 2018b*) available from https://www.github.com/4ment/physher.

## METHODS

We have implemented phylogenetic models in a Python software package called `phylostan`. `phylostan` accesses the Stan package through the pystan API (*Stan Development Team, 2019*). When executed, the `phylostan` command line program generates and runs a Stan script with the selected phylogenetic model. Table 1 shows a summary of different types of models available in `phylostan`. The code and the associated datasets are available from https://www.github.com/4ment/phylostan.

### Variational inference

The main idea behind variational inference is to transform posterior approximation into an optimization problem using a family of approximate densities. The aim is to find the member of that family with the minimum KL divergence to the posterior distribution of interest:

$$\phi^* = \arg\min_{\phi \in \Phi} \mathrm{KL}(q(\theta; \phi) \| p(\theta \mid D, \tau)),$$

where $q(\theta; \phi)$ is the variational distribution parameterized by a vector $\phi \in \Phi$, while $\theta$ are the model parameters (e.g. branch lengths), $D$ are the aligned sequence data, $\tau$ is a fixed tree topology, and KL is defined as

$$\mathrm{KL}(q\|p) = \int_{\theta} q(\theta; \phi) \log \frac{q(\theta; \phi)}{p(\theta \mid D, \tau)}.$$

To minimize the KL divergence, we first rewrite the KL equation:

$$\mathrm{KL}(q(\theta; \phi) \| p(\theta \mid D, \tau) = \mathbb{E}[\log q(\theta; \phi)] - \mathbb{E}[\log p(\theta \mid D, \tau)]$$
$$= \mathbb{E}[\log q(\theta; \phi)] - \mathbb{E}[\log p(\theta, D \mid \tau)] + \log p(D \mid \tau),$$

where the expectations are taken with respect to the variational distribution $q$. The third term $\log p(D \mid \tau)$ on the right hand side of the last equality is a constant with respect to the variational distribution so it can be ignored for the purpose of the minimization. After switching the sign of the other two terms, the minimization problem can be framed as a maximization problem of the evidence lower bound (ELBO) function

$$\mathrm{ELBO}(\phi) = \mathbb{E}[\log p(\theta, D \mid \tau)] - \mathbb{E}[\log q(\theta; \phi)].$$

The ELBO is easier to calculate than the KL divergence as it does not involve computing the intractable posterior nnormalization term $p(D \mid \tau)$. We entertained two classes of variational distributions: mean-field and full-rank Gaussian distributions. The mean-field approximation assumes a complete factorization of the distribution over each of the $N$ parameters of the model (e.g. branch lengths, GTR parameters) and each factor is governed by its own variational parameters $\phi_i$:

$$q(\theta_1, \ldots, \theta_N; \phi) = \prod_{i=1}^{N} q(\theta_i; \phi_i),$$

where $q(\theta_i; \phi_i)$ is a Gaussian density and $\phi_i = (\mu_i, \sigma_i)$.

The full-rank approximation is

$$q(\theta_1, \ldots, \theta_N; \phi) = \mathcal{N}(\theta; \phi),$$

where the term $\phi = (\mu, \Sigma)$ concatenates the mean vector $\mu$ and covariance matrix $\Sigma$ of a multivariate Gaussian distribution. The mean vector and the covariance matrix contribute respectively $N$ and $N(N + 1)/2$ parameters to the full-rank variational model.

When provided with a model definition via `phylostan`, the Stan software is able to estimate the variational parameters by applying stochastic gradient ascent in a black box approach (*Ranganath, Gerrish & Blei, 2014*; *Kucukelbir et al., 2015*).

In phylogenetics, parameters are usually constrained, for example, branch lengths and substitution rates are non-negative while nucleotide frequencies sum to one. Since the support of the Gaussian distribution is $\mathbb{R}$, Stan automatically transforms constrained parameters into unconstrained variables. For example, branch lengths are log-transformed such that they live in the real coordinate space. Stan maintains a set of transformations for common constraints (e.g. lower and/or upper bound constraints) and their corresponding Jacobians. Further information on the library of transformations can be found in the online Stan manual https://mc-stan.org/docs/2_18/reference-manual/variable-transforms-chapter.html.

## Models with clocks

The simplest phylogenetic models do not distinguish between evolutionary rate and time, and for those models the only constraint on the branch lengths of a tree is that they must be non-negative. Since our models assume that branches are independent, the non-negativity is easily accommodated in a Stan model via Stan's application of a log transform on the branch length parameters.

On the other hand, trees in molecular clock models are constrained so that parent nodes must be older than any of their descendent nodes. Here we consider that time is going backwards: the earliest taxon was sampled at time 0 and the age of every internal node is greater than 0. In the case of homochronous sequence data (i.e. every sequence was collected at the same time), the age of an internal node can be reparameterized as a proportion of the age of its parent, except for the root age. For each internal node $i$ (excluding the root), the height $h_i$ of the node is reparameterized as $h_i = p_i \, h_{pa(i)}$ where the subscript $pa(i)$ denotes the parent of node $i$ and $0 < p_i < 1$. The height of the root is constrained to be non-negative.

In the case of heterochronous sequence data, the age of the sequence data at the tips of the tree must to be taken into account. For each internal node $i$ (excluding the root), the height $h_i$ of the node can be reparameterized as $h_i = h_{d(i)} + p_i (h_{pa(i)} - h_{d(i)})$ where the subscript $d(i)$ denotes the earliest taxon in the set of descendant nodes of node $i$. The height of the root is now constrained to be greater than the earliest taxa.

The reparameterized model consists of a vector of proportions $p = (p_1, \ldots, p_{N-2})$, which Stan automatically transforms using a log-odds transform.

This reparametrization of the node heights is also used in the PAML software package (*Yang, 2007*). In our case, the transformation of the node ages requires an adjustment to the joint density. Specifically, in the multivariate case, the density is scaled by the absolute value of the Jacobian of the inverse of the transforms. The Jacobian of homochronous data is a triangular matrix for which the determinant is $\prod_{i=1}^{N-2} h_{pa(i)}$. For heterochronous data, the Jacobian is also triangular for which the determinant is $\prod_{i=1}^{N-2} h_{pa(i)} - h_{d(i)}$.

Under these reparameterizations, different ranked topologies will be evaluated during the gradient ascent optimization of the variational parameters.

## Rate heterogeneity among sites

The gamma distribution is commonly used to model rate heterogeneity across sites in phylogenetic models (*Yang, 1994*, *1996*). However, it is currently not possible to use the quantile approximation of the gamma distribution proposed by *Yang (1994)* since the Stan language does not provide inverse cumulative density functions (CDF). Instead we opted to use the Weibull distribution to describe rate heterogeneity across sites. The Weibull distribution has the closed-form inverse CDF $F^{-1}(p \mid \lambda, k) = \lambda (-\ln(1 - p))^{1/k}$. Fixing the scale parameter $\lambda$ to 1, the shape of the Weibull distribution is determined by the shape parameter $k$. When $k \leq 1$ the Weibull distribution is skewed, suggesting strong rate heterogeneity. When $k > 1$ the distribution is bell shaped, suggesting no or low rate heterogeneity. Since the mean of the Weibull distribution must be equal to 1 in order to preserve branch length interpretation (i.e. expected number of substitutions per site) we rescale the rates $r_1 \ldots r_K$ such that $\sum_{i=1}^{K} p_i r_i = 1$. Such rescaling is also necessary when a proportion of invariant sites is included in the model.

We also implemented rate heterogeneity across sites using a general discrete distribution with $2K - 2$ parameters, by specifying $K$ rates $0 \leq r_1 < r_2 \ldots < r_K$ and probabilities $p_i = Pr\{r = r_i\}$ with $\sum_{i=1}^{K} p_i r_i = 1$ and $\sum_{i=1}^{K} p_i = 1$ (*Kosakovsky Pond & Frost, 2005*). The values of $r_i$ are constrained to be ordered from smallest to largest merely for the sake of model identifiability in Stan. This distribution is parameter rich compared to the discretized Weibull distribution scheme and as a result it may be more difficult to fit to data.

## Datasets and validation

To evaluate the accuracy of variational inference we applied `phylostan` to three empirical datasets and compared the posterior approximations to those obtained with BEAST2 (*Bouckaert et al., 2019*), BEAST (*Suchard et al., 2018*), or MrBayes (*Ronquist et al., 2012*). The first dataset is made of 69 human influenza virus haemagglutinin nucleotide sequences

isolated between 1981 and 1998. The dataset was analyzed under the Hasegawa, Kishino and Yano (HKY) substitution model allowing for rate variation among sites (Weibull distribution with four rate classes). We used a strict clock and a constant size coalescent prior on the tree. The topology used in the `phylostan` and BEAST2 analyses was drawn randomly from a sample of trees generated by a preliminary analysis with BEAST2 without topological constraints.

The second dataset comprises 63 RNA sequences of type 4 from the E1 region of the hepatitis C virus (HCV) genome that were isolated in 1993. As in previous studies (*Pybus et al., 2001*; *Faulkner et al., 2018*), the substitution rate was fixed to $7.9 \times 10^{-4}$ substitutions per site per year. We used the GTR substitution model with Weibull distributed rate heterogeneity (four categories). We used the Bayesian skyride tree prior and a Gaussian Markov random field prior for the effective population size trajectory (*Minin, Bloomquist & Suchard, 2008*). The topology used in the `phylostan` and BEAST (*Suchard et al., 2018*) analyses was also drawn randomly from a sample of trees generated by a preliminary analysis with BEAST without topological constraints. We assigned a gamma prior with rate and scale equal to 0.005 on the precision parameter.

The last dataset consists of twenty-seven 18S rRNA sequences of length 1949. This dataset, commonly referred to as DS1, has been studied several times and has become a *de facto* standard dataset for evaluating MCMC methods (*Hedges, Moberg & Maxson, 1990*; *Lakner et al., 2008*; *Fourment et al., 2018b*; Whidden and Matsen IV, 2015; *Whidden et al., 2018*). DS1 was analyzed under the Jukes Cantor substitution model (JC69) without a clock. In (*Fourment et al., 2018b*), the authors used very long runs of MrBayes (*Ronquist et al., 2012*) to estimate the posterior distribution of the tree topologies (see corresponding paper for more details). Akin to that study, we focused on the 42 tree topologies contained in the 95% credible set and renormalized their posterior probabilities so that they sum to one. For each topology we then estimated its ELBO under the mean-field approximation using either Stan or `physher` (*Fourment & Holmes, 2014*), and converted these approximations of the marginal likelihood to a posterior probability. Specifically, we use the ELBO estimate $\hat{p}_{\mathrm{ELBO}}(D \mid \tau) := \max_{\phi \in \Phi} \mathrm{ELBO}(\phi)$ to approximate the marginal likelihood of a topology. Finally, we compare the ELBO-based posterior estimates to the MCMC-based estimates from MrBayes, which we consider as the ground truth. We placed independent exponential priors on branch lengths with a prior expectation of 0.1 substitutions.

In every analysis the gradient of the ELBO is evaluated with one sample and the ELBO is computed using 100 samples. While we used the default parameters provided by pystan, we used a smaller relative tolerance of 0.001 for every analysis as suggested by *Yao et al. (2018)*.

## RESULTS

We analyzed three datasets to evaluate the accuracy of the approximations to the posterior distribution provided by the variational framework on phylogenetic models. First, we analyzed a set of heterochronous influenza A virus sequences under the strict clock model on a fixed topology with BEAST2 and `phylostan`. In Fig. 1 we show the posterior

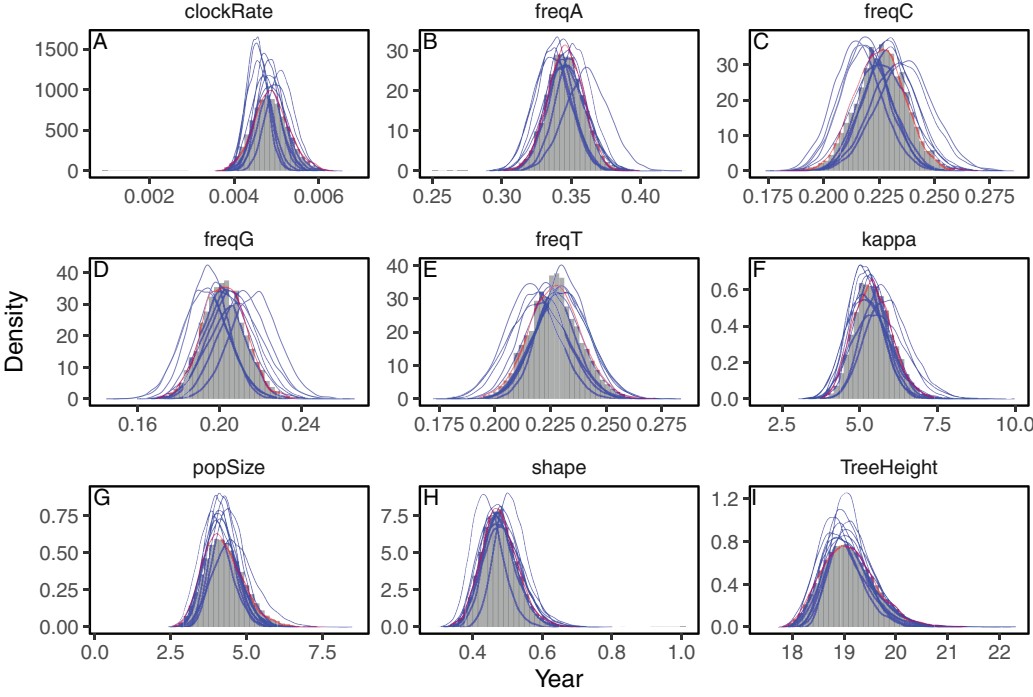

**Figure 1 Posterior approximation of phylogenetic model parameters using mean-field variational inference (phylostan), NUTS (phylostan), and Metropolis-Hastings (BEAST2) algorithms on the influenza A virus dataset.** Mean-field variational inference (blue lines) was replicated 10 times while the NUTS (red line) and the Metropolis-Hastings (histogram) methods were run once. Parameters are: clock substitution rate (A), nucleotide frequencies (B–E), ratio of transition and transversion rates of HKY model (F), population size (G), shape parameter of the Weibull distribution (H), and tree height (I).

distributions of every parameter approximated by BEAST2 and phylostan using either mean-field variational inference or the NUTS algorithm. Although the mean-field-based variational approach tends to approximate the posterior distributions correctly, the between-run variability is high relative to other methods. Apart from two very divergent replicates, the full-rank variational approximation appears to be more precise than the mean-field approximation (Fig. 2). The two replicates that do not correctly approximate the posterior distributions have a substantially lower ELBO (−4475.75 and −4481.07) than the other replicates for which the ELBO are between −4430 and −4432, so in practice these poorly converged runs could be identified and removed by comparing the results of multiple replicates. The distributions inferred by BEAST2 using Metropolis Hastings closely match the posterior approximations given by the NUTS algorithm implemented in Stan, providing evidence that we have implemented the models in phylostan correctly.

Next we evaluated the ability of the variational framework to approximate the demographic history of a set a HCV using the skyride model (*Minin, Bloomquist & Suchard, 2008*) and a fixed substitution rate. In Fig. 3, we present the demographic history approximations from the replicate with the highest ELBO for the mean-field and full-rank models. The trajectories of the population size estimated from the variational distributions match the estimates from BEAST and NUTS (Fig. 3). The 95% confidence

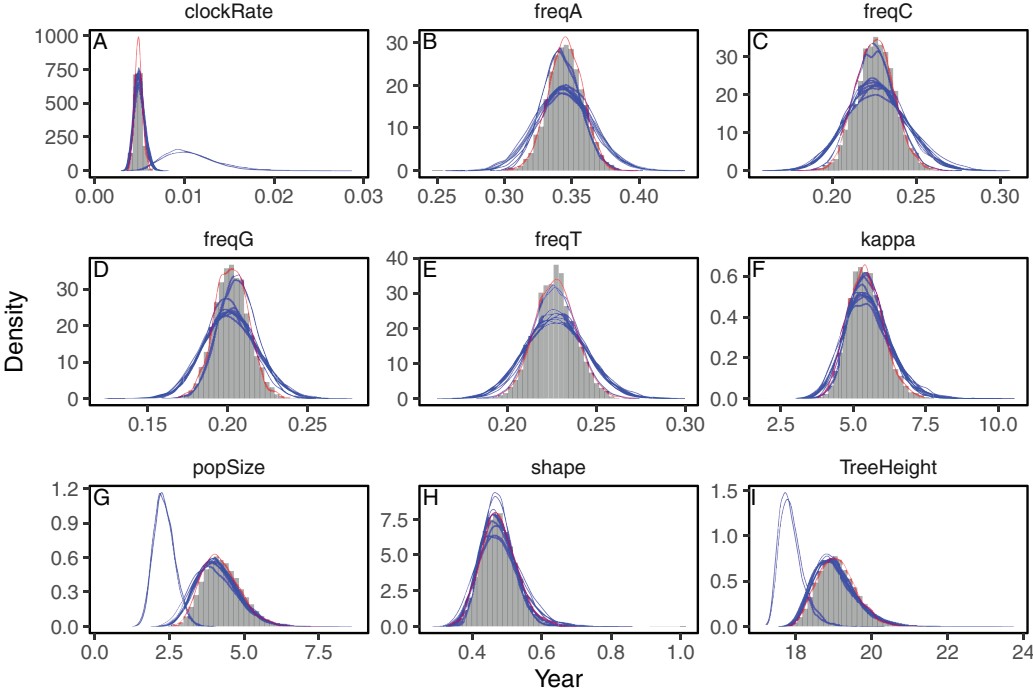

**Figure 2 Posterior approximation of phylogenetic model parameters using full-rank variational inference (phylostan), NUTS (phylostan), and Metropolis-Hastings (BEAST2) algorithms on the influenza A virus dataset.** Full-rank variational inference (blue lines) was replicated 10 times while the NUTS (red line) and the Metropolis-Hastings (histogram) methods were run once. Parameters are: clock substitution rate (A), nucleotide frequencies (B–E), ratio of transition and transversion rates of HKY model (F), population size (G), shape parameter of the Weibull distribution (H), and tree height (I).              

intervals computed from the variational distributions are narrower than the MCMC-based intervals. This can, at least in part, be explained by the zero-forcing nature of the reverse KL divergence. Although the mean-field variational approximation consistently recovers the population size trajectory across replicates, the full-rank approximation fails to converge to the correct posterior in 9 out of the 10 replicates.

In the final analysis we compare the accuracy and speed of mean-field variational inference as implemented in `phylostan` against an implementation in the `physher` software when using the Jukes–Cantor model. `phylostan` and `physher` use the same gradient ascent algorithm and variational distribution, for which the gradient is analytically derived (*Kucukelbir et al., 2015*). We find that `physher` is two orders of magnitude faster than `phylostan` with a mean computing time of 1.2 s per topology for the former and 332 seconds per topology for the latter (Fig. 4). Although it is much faster, the `physher` implementation of mean-field variational inference currently is specific to the Jukes–Cantor model and lacks the generality of `phylostan` to carry out inference under a range of models. Another important aspect that may contribute to the speed differences is how each program initializes its variational parameters. Whereas Stan initializes the variational parameters to random points in the parameter space, `physher` initializes these parameters deterministically with near optimal values using the

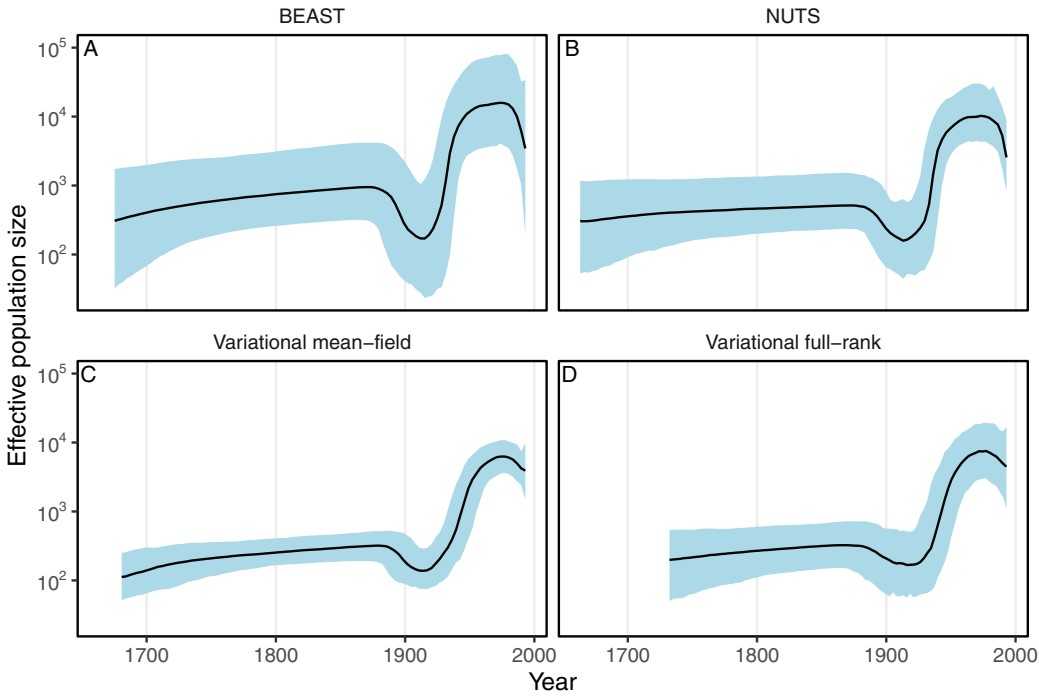

**Figure 3 Posterior medians (solid black lines) of effective population sizes and associated 95% credible intervals (blue shaded areas) for the HCV dataset using BEAST (A) and** `phylostan` **(B–D).** Variational-based approximations shown in this figure are from the replicates with the highest ELBO. Variational-based approximations for every replicate can found in Figs. S1 and S2.

Laplus method (*Fourment et al., 2018b*). For each branch length parameter, the Laplus approximation takes its maximum *a posteriori* estimate and second derivative with respect to the log-likelihood and finds the parameters (μ and σ) of the log-normal distribution by matching modes and second derivatives of the approximating and posterior distributions of the branch length. The procedure is described in more detail in (*Fourment et al., 2018b*). Since we use a log-transform on the branch length parameters, the two parameters $\mu$ and $\sigma$ are used to initialize the location and scale parameters of the normal variational distribution.

## DISCUSSION

We have developed a tool based on the Stan package for Bayesian phylogenetic inference, which to our knowledge is the first application of VB to time trees with coalescent models. Although we have focused on inferring phylogenetic models with a fixed topology due to the complexity and discrete nature of the topology space, recent research on subsplit Bayesian networks (SBN) has made a significant step toward modeling topological uncertainty in the variational framework (*Zhang & Matsen, 2018*, *2019*). A promising area for future research will be to investigate combinations of the parameter rich models we have presented in this study with the SBN approach, for which only the JC69 model has been implemented thus far.

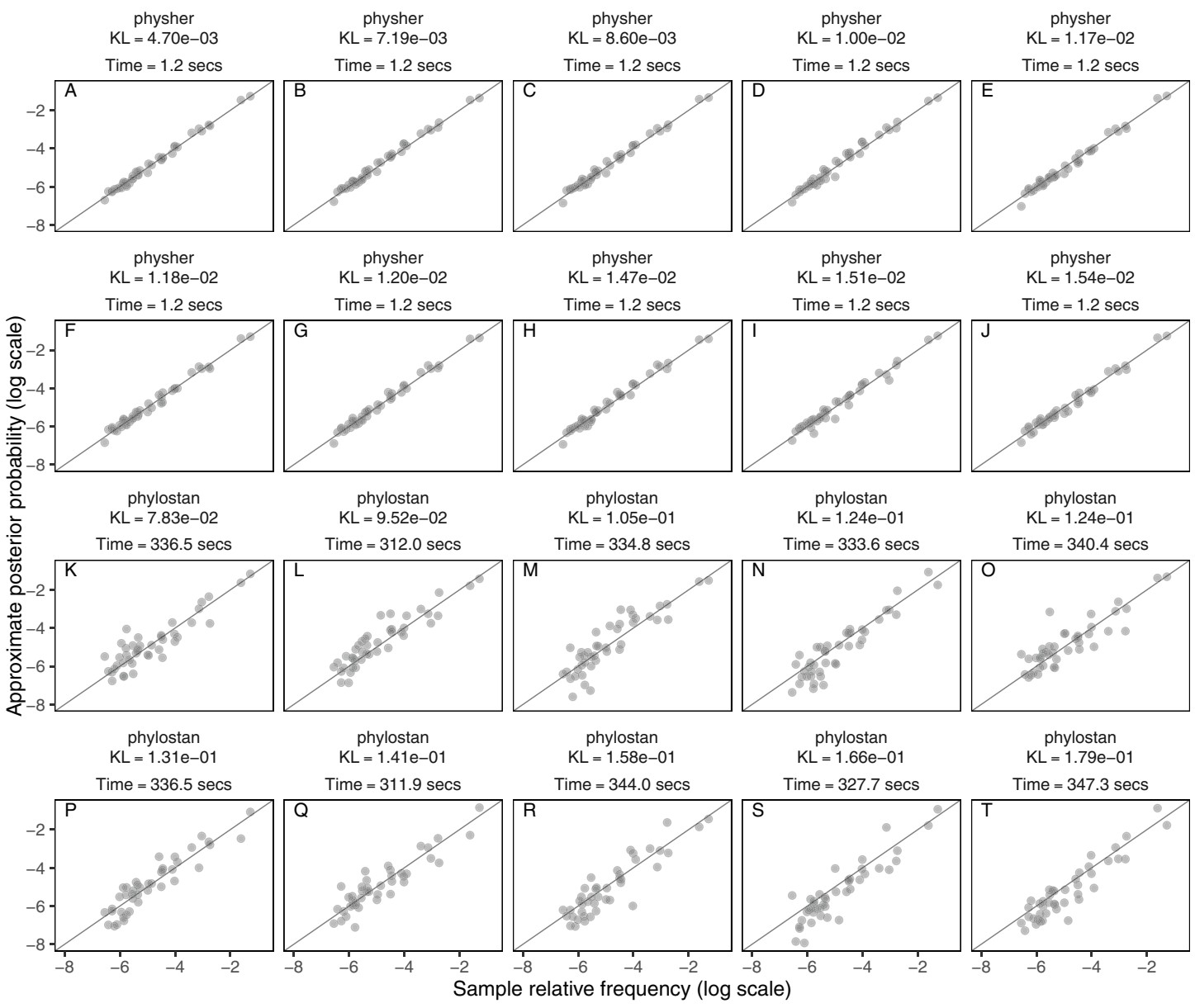

**Figure 4 The approximate posterior probabilities of the topologies in DS1 (y-axis) calculated with physher (A–J) and phylostan (K–T) versus the ground truth posterior probabilities from MrBayes (x-axis), plotted on the log scale for clarity.** The mean time per topology and the Kullback–Leibler divergence are reported for each of the 10 replicates. Panels are ordered in decreasing order by Kullback–Leibler divergence.

In our tests, we found the variational inference engine of Stan to be considerably faster than the NUTS algorithm. We then showed using the DS1 dataset that a manually coded implementation of variational inference can be two orders of magnitude faster and somewhat more accurate than the automatic differentiation variational inference produced via Stan. This finding makes it clear that the modeling flexibility and ease of use afforded by a general purpose language like Stan comes with significant tradeoffs in computational efficiency and accuracy. On the other hand, bespoke implementations of

variational inference can be complex to derive, time consuming to implement, and limited in flexibility. For example, the Laplus method used to initialize the parameters of the branch length variational distributions assumes that the branch length parameters are independent. Further work would be needed to extend the Laplus method for other model parameters that are strongly correlated such as the GTR parameters. Opportunities may exist to combine the flexibility of a general purpose modeling language with an inference engine that is specialized for phylogenetic models. Implementation of variational inference within widely used phylogenetic packages such as BEAST, MrBayes, and RevBayes (*Höhna et al., 2016*) would be a significant step forward. Although calculating the gradient of each parameter analytically is in theory possible, this tedious task could be circumvented by adopting the automatic differentiation framework used in modern probabilistic programming languages (e.g. Stan, Edward).

In this study we have presented and evaluated a subset of the functionality available in `phylostan`. In addition to the models evaluated here we have also implemented the coalescent skygrid model (*Gill et al., 2013*) and more flexible molecular clock models such the relaxed clock using a hierarchical prior and an autocorrelated clock rate. A list of model features available in `phylostan` is given in Table 1.

Another promising avenue for future research would be to capture dependencies between latent variables that, by definition, are ignored by the mean-field approximation. To this end, *Tran, Blei & Airoldi (2015)* have proposed to augment the mean-field variational distribution with a copula, while recent studies (*Rezende & Mohamed, 2015*; *Kingma et al., 2016*) have proposed to improve the posterior approximation though a normalizing flow. Although the full-rank approximation can in principle capture some of these dependencies, in practice it seems that phylogenetic models can not be reliably fit with the full-rank approximation, at least not with the black box approach implemented in Stan. We speculate that the failure to fit the full-rank approximation on, e.g. the HCV dataset shown above, is a result of the high dimension of the covariance matrix. One possible remedy would be the use of a sparse covariance matrix to reduce the number of parameters. It is reasonable to assume that some variables have low or no correlation (e.g. guanine frequency and root height, or pairs of branch lengths for distant tree branches). It may also be possible to apply a low-rank Gaussian distribution (*Miller, Foti & Adams, 2017*) to alleviate the computational burden associated with the full-rank distribution.

## CONCLUSIONS

Although the methods we have introduced are limited to analysis of a single fixed tree topology, and have much room for improvement, there is still scope for them to be usefully applied to biological data analysis. Phylogenetic methods have proven especially useful in the context of infectious disease research. In particular, systems such as Nextstrain (*Hadfield et al., 2018*) provide a fast way to incorporate data into a visualization environment that enables exploration of phylogeny and other key aspects such as sample collection time and geographical location. Nextstrain computes a phylogeny for input

sequences using maximum likelihood-based methods. In order to reconstruct temporal information, it then takes the single maximum likelihood tree and applies the TreeTime software (*Sagulenko, Puller & Neher, 2018*) to infer divergence times and evolutionary rates. TreeTime provides a fast and highly scalable means to carry out approximate maximum likelihood inference on those model parameters. Although the current implementation may not run as fast as TreeTime, `phylostan` could offer a drop-in replacement to something like TreeTime in this application context. `phylostan` would have the advantage of carrying out joint Bayesian inference of all continuous model parameters, and being able to report the uncertainty associated with estimates of each model parameter.

## ACKNOWLEDGEMENTS

We would like to thank Erick Matsen, Alexei Drummond, Christiaan Swanepoel and Cheng Zhang for useful discussions.

### Funding

This work was supported by the ithree institute, UTS and AusGEM. The funders had no role in study design, data collection and analysis, decision to publish, or preparation of the manuscript.

### Grant Disclosures

The following grant information was disclosed by the authors:
ithree institute, UTS and AusGEM.

### Competing Interests

The authors declare that they have no competing interests.

### Author Contributions

- Mathieu Fourment conceived and designed the experiments, performed the experiments, analyzed the data, contributed reagents/materials/analysis tools, prepared figures and/or tables, authored or reviewed drafts of the paper, approved the final draft.
- Aaron E. Darling conceived and designed the experiments, contributed reagents/ materials/analysis tools, approved the final draft.

### Data Availability

Data is available from GitHub: https://github.com/4ment/phylostan.

### Supplemental Information

Supplemental information for this article can be found online at http://dx.doi.org/10.7717/peerj.8272#supplemental-information.

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
