# Peer review of "Evaluating probabilistic programming and fast variational Bayesian inference in phylogenetics"

_PeerJ, doi:10.7717/peerj.8272_

## Round 0.1 · original submission · Minor Revisions

The reviewers provided concrete comments that should be addressed in your resubmission - see details below.

Reviewer 1 ·

Basic reporting

This manuscript describes the first application of variational inference for Bayesian phylogenetics. This work is quite pioneer work, which has the potential to have a strong impact in the phylogenetics comunity. The manuscript is well written and easy to follow, in spite of the difficulty of this topic.

Experimental design

The methods were carefully designed and relevant to the aims of this study. However, there are a few points that are unclear and potentially important to assess the utility of variational inference:

1- Fig 3 shows effective population size reconstructions using four methods. That from BEAST has higher uncertainty. This could be due to uncertainty in the tree topology, which is not accounted for in variational inference. Can the authors confirm whether the tree topology estimated in BEAST, or fixed as in the other methods. If it is the latter, is there a potential for biassed estimates using variational inference? How does this affect run time?

2- Fig 2 shows the posterior estimates of several parameters. A couple of variational inference replicates were different. I think some guidelines are missing here. For example,
how many times should one run variational inference? Should one take the mode or a an average?

3- I realise that an extensive comparison of methods to approximate the posterior distribution is beyond the scope of this study, but I do think readers would appreciate some discussion and comparisons with sequential Monte Carlo, which is an other alternative to MCMC that might be more amenable to large data sets.

4- The difference in computing time between physher and stan is over three orders of magnitude, and it has been attributed to the starting values, where physher uses an optimisation step whereas stan has random starting points. Would this be a problem for parameters with local optima or for those that are unidentifable? For example, when calibration data are not very informative, the posterior on root height and clock rate are rather flat and difficult to sample using MCMC.

Validity of the findings

The findings are easily replicated and the authors have made substantial effort to document their code and examples in GitHub.

Additional comments

I am happy to recommend this manuscript for publication after my points in section 2 are addressed.

Reviewer 2 ·

Basic reporting

The paper is on fixed topology inference, but the introduction mentions NNI and SPR moves before backing down later. A reference to fixed topologies in the abstract would not bring up hopes, and would more accurately reflect the contents of the paper.

line 34: "etc" => "etc."

lines 108-109 suggest Gaussian distributions are used for GTR rates, branch lengths, but they are log transformed version of it, right? Otherwise, there will be support for negative branch lengths and GTR rates.

Figure 3 could be improved by either plotting all trajectories on top of each other, or at least add a grid across all backgrounds so that different trajectories can be compared more easily.

Figure 4: axis represent *log* posterior probabilities?

An omission is the way the tree priors are described -- since these are coalescent priors and even on a fixed topology suffer from discrete jumps. The trees ((a:1,b:1):2,(c:2,d:2):1) and ((a:2,b:2):1,(c:1,d:1):2) are two differently ranked topologies, so how does the tree prior, in particularly the derivatives deal with these?

A description of the parameterisation (if any) for frequencies and substitution model rates would be nice to have as well.

line 136 "requires an adjustment to the joint density" how exactly?

line 190: missing reference.

Experimental design

Line 58 promises "1000s of sequences", but the largest data set in the experiments is only 69 sequences. Please add a more substantial data set.

A coverage experiment (draw 100 parameter + tree sets from prior, count how often the true values are inside the 95%HPD of the variational distribution) would provide much more insight in the behaviour of the methods. Anecdotal evidence from a handful of datasets is interesting, but not as illuminating.

Validity of the findings

Results are hard to replicate due to missing BEAST XML files, alignment files, phylostan and physher runtime settings. Please provide these.

It is unclear from the documentation how to run physher for variational inference (assuming I can find it at https://github.com/4ment/physher, which was not clear from the manuscript, and the referred paper only mentions a google code archive). Please update the documentation.

How do we know variational inference in physher is correctly implemented? There is no evidence provided as for phylostan (through Figures 1 & 2). Adding these would make the results more credible.

A performance comparison with Zhang & Matsen (2019) would greatly enhance the paper.

Additional comments

line 91: Trees tend to contain the majority of the parameters: for n taxa, there are 2n-1 parameters for the topology (parents for all nodes) + n-1 heights of internal nodes + potentially n heights for tip nodes (if estimated). The number of parameters for the site model tends to be tiny compared to that of the tree, which suggests solving site model inference is just a small part of the solution. It would be good if you substantiate the claim that "Despite this limitation, implementing the type of model we describe here under a fixed tree is a useful step forward toward fast Bayesian in-ference of complex phylogenetic models, as it helps us understand the quality and speed of posterior approximation that can be achieved using a generic modelling language like Stan".



lines 243 ff: I find the discussion a bit confusing: the line seems to be that stan has significant costs, justifying implementations in other packages, which admittedly can be tedious because it requires gradients, which should be obtained from stan. Should we use stan or not? Can the improvement of the phylostan be attributed to more efficient gradient implementations or other parts? How do you know?

---

## Round 0.2 · accepted · Accept

The Reviewer comments have been addressed appropriately.